# The Role of ALD-ZnO Seed Layers in the Growth of ZnO Nanorods for Hydrogen Sensing

**DOI:** 10.3390/mi10070491

**Published:** 2019-07-23

**Authors:** Yangming Lu, Chiafen Hsieh, Guanci Su

**Affiliations:** Department of Electrical Engineering, National University of Tainan, Tainan 70005, Taiwan

**Keywords:** ALD, ZnO, nanorods, hydrothermal process, hydrogen sensor

## Abstract

Hydrogen is one of the most important clean energy sources of the future. Because of its flammability, explosiveness, and flammability, it is important to develop a highly sensitive hydrogen sensor. Among many gas sensing materials, zinc oxide has excellent sensing properties and is therefore attracting attention. Effectively reducing the resistance of sensing materials and increasing the surface area of materials is an important issue to increase the sensitivity of gas sensing. Zinc oxide seed layers were prepared by atomic layer deposition (ALD) to facilitate the subsequent hydrothermal growth of ZnO nanorods. The nanorods are used as highly sensitive materials for sensing hydrogen due to their inherent properties as oxide semiconductors and their very high surface areas. The low resistance value of ALD-ZnO helps to transport electrons when sensing hydrogen gas and improves the sensitivity of hydrogen sensors. The large surface area of ZnO nanorods also provides lots of sites of gas adsorption which also increases the sensitivity of the hydrogen sensor. Our experimental results show that perfect crystallinity helped to reduce the electrical resistance of ALD-ZnO films. High areal nucleation density and sufficient inter-rod space were determining factors for efficient hydrogen sensing. The sensitivity increased with increasing hydrogen temperature, from 1.03 at 225 °C, to 1.32 at 380 °C after sensing 100 s in 10,000 ppm of hydrogen. We discuss in detail the properties of electrical conductivity, point defects, and crystal quality of ALD-ZnO films and their probable effects on the sensitivity of hydrogen sensing.

## 1. Introduction

Hydrogen is an important potential source of clean energy for the future. However, its explosiveness necessitates specific safety measures, so developing high-sensitivity hydrogen sensors is a high priority. The metal oxide semiconductor type sensor has the characteristics of low cost, easy preparation, long life, high sensitivity, short reaction time, and small volume, and thus has become the most common and practical gas sensor at present. Gas sensors made from metal oxide semiconductors are referred to as resistive gas sensors, or chemiresistors, and they respond with resistance changes when exposed to reducing gas atmospheres (e.g., H_2_, C_2_H_5_OH) or oxidizing gas atmospheres (e.g., CO, O_2_). It is generally believed that sensing in chemiresistors involves either the grain boundary barrier or the surface conductivity mode [1,2]. The gas sensing mechanism can be illustrated by taking the N-type semiconductor material, ZnO, as an example. When the ZnO gas sensing material is placed under normal atmospheric conditions, O_2_ in the air will adsorb to the surface of ZnO. The adsorbed O_2_ will extract an electron from ZnO surface and transform itself into an O^2−^ state, resulting in a decrease in electron density on the ZnO surface and an increase in resistance. When slightly heated, it accelerates the adsorption of O_2_ on the surface of ZnO to form a more stable adsorption state. If the temperature is higher than 150 °C, the O^2−^ will convert to an O^−^ adsorption state with electron. When the N-type semiconductor material contacts with the atmosphere of a reducing gas, the O^−^ originally adsorbed on the ZnO surface will react with the reducing gas and release electrons back to the ZnO. This will increase the electron density of the ZnO and lead to a decrease in the resistance of it [3]. Commonly used metal oxides for resistive hydrogen sensing are SnO_2_ [4], TiO_2_ [5], WO_3_ [6,7], and ZnO [8]. Metal oxides are attractive for their straightforward manufacturing, low cost, good reactivity, and simple circuit structure. Tin oxide (SnO_2_), which was the first to be developed and applied, has the advantages of high sensitivity and short reaction time [9]. Unfortunately, it is easily oxidized by water vapor, has low conductivity, and requires higher operating temperatures (400–500 °C). Compared to tin oxide, zinc oxide has higher carrier mobility (200 cm^2^/V·s) [10], lower cost, and a lower operating temperature range (25–400 °C) [11]. Hence, ZnO is more suitable as a gas sensing material than SnO_2_. Basu and Dutta [12] developed two kinds of sensor structures, viz., a Pd/ZnO/p-Si heterojunction and Pd/ZnO/Zn metal-active insulator-metal (MIM) for hydrogen sensor applications. These two structures mainly rely on the hydrogen sensing properties of the precious Pd metal itself, not the individual contribution of the ZnO material. Phanichphant [13] once reviewed the use of oxide semiconductors as a hydrogen sensor in 2014. From his review, it can be found that almost all of the materials are based on WO_3_ materials and all need to add precious metal Pt or Pd to get good responses. Without the help of these precious metals, it is necessary to detect hydrogen at temperatures of 200–300 °C or more.

In our research, cheap, non-toxic and abundant zinc oxide materials were used. It can detect low concentrations of hydrogen without the help of expensive rare metals. The most important reason for this is that we had prepared the structure of the zinc oxide nanorods and improved the material properties and electrical properties by the interface material. In other words, the introduction of ALD-ZnO seed layer. Nanostructures of ZnO have attracted considerable research attention because they present a wide variety of morphologies with large surface areas, including: One-dimensional nanorods, nanotubes, and nanowires; two-dimensional nanowalls; three-dimensional nanoflower and “sea urchin” shapes; and spheres [14,15,16,17,18,19,20,21]. Gas sensors using nanostructures have better physical and chemical properties than thin films because of the former’s larger surface-to-volume ratios, which lead to larger reaction areas, enhanced sensitivity, and shorter response times.

## 2. Experimental Details

We used atomic layer deposition (ALD) to deposit ZnO thin films, alternately injecting diethylzinc (DEZn) and water vapor as precursor reactants over a growth temperature range of 70–150 °C. The ALD-ZnO thin films subsequently served as seed layers to grow ZnO nanostructures via a hydrothermal process. Figure 1 represents the sequence and timing of the ALD process, while Figure 2 presents a schematic of the ALD-ZnO film growth process. The overall growth of ALD-ZnO cycle is mainly divided into four steps. Firstly, DEZn is introduced, which reacts with the hydroxyl group on the surface of the substrate to form a zinc-oxygen bond and produces an ethane (C_2_H_6_) by-product. In the second step, argon (Ar) is introduced to carry away the excess ethane (C_2_H_6_) by-product molecules and unreacted DEZn molecules. In the third step, water molecules are introduced to provide hydroxyls to bond zinc atoms and some ethane (C_2_H_6_) by-products are formed. In the fourth step, argon (Ar) is again introduced to carry away residual water molecules and excess ethane (C_2_H_6_) by-products molecules. The times to inject DEZn, argon, water, and argon in sequence are set to 1 s, 5 s, 1 s, and 5 s on an automatic controller. ZnO nanorods were grown hydrothermally, using a water bath preheated to 90 °C for 60 min before the hydrothermal reaction was carried out. Solutions of 0.3 M zinc nitrate and 0.3 M hexamethylenetetramine (HMTA) were homogeneously mixed in a 1:1 volume ratio (45 mL:45 mL). The hydrothermal reaction was maintained at 90 °C for 9 h. A Si substrate with a ALD-ZnO seed layer was fixed on a bracket and dropped into the solution vessel, then the vessel mouth was completely sealed before the hydrothermal reaction was begun. The resulting ZnO nanostructures were structurally analyzed using X-ray diffraction (XRD) with Cu Kα (0.1542 nm) radiation. The surface morphology was observed with a high-resolution scanning electron microscope (SEM; Hitachi SU8000, Tokyo, Japan). Room-temperature photoluminescence (PL) spectra were obtained using a Jobin Yvon spectrometer with a HeCd laser (325 nm, 20 mW).

For the prepared zinc oxide nanomaterial, the measurement of hydrogen gas sensing characteristics is carried out. The volume percentage of nitrogen 79% and oxygen 21% is used to simulate the general atmospheric state. Hydrogen is stored in different gas cylinders by diluting to 10,000 ppm with nitrogen. During the measurement, the diluted hydrogen gas is injected into the sealed stainless steel sensing chamber.

## 3. Results and Discussion

In an ALD growth system, there is a temperature interval during which the film’s growth rate remains constant, called the ALD growth window. Figure 3 shows that the ALD growth window in this study extended from 100 to 125 °C. When the temperature was below 100 °C, there was insufficient energy, so the growth rate was very low. When the temperature exceeded 125 °C, the growth rate rose. According to the literature [22], a higher growth temperature will result in less OH^−^ being dissociated from H_2_O precursors. Under such conditions, the reaction between diethylzinc and OH^−^ radicals yields Zn-Zn bonds rather than atomic layers of ZnO. In this study, the ALD-seed layers were prepared at 125 °C with thickness of 100 nm.

In general, PL is used to assess crystal quality and point defects in ZnO. The contribution of amorphous ZnO to the intensity of the PL spectrum is very small. However, PL intensity can be enhanced by the formation of crystalline structures, especially in low-dimensional structures [23]. Theoretically, the PL spectra of ZnO films or ZnO nanorods have two major components: One is due to near-band-edge emission originating from an exciton transition, while the other is a broad visible band due to defect emissions arising from structural defects or impurities [24,25]. The PL spectrum of ZnO has a significant peak at around 387.5 nm in the UV region (Figure 4), which arises from exciton emissions and is a good indicator of the quality of a particular ZnO thin film [26,27]. It can be seen from Figure 4 that ALD-ZnO had good crystallinity even at temperatures as low as 70 °C. Near 580 nm, there is a broader yellow-green peak. According to the literature [28], this peak is due to oxygen vacancies, as is another peak at 667.5 nm. The results in Figure 4 show that whereas the peak at 387.5 nm increased as the temperature rose, the peaks at 580 and 667.5 nm gradually decreased. Oxygen vacancy defects in the crystal decreased as the growth temperature rose, indicating that high-quality ZnO films with fewer point defects could be obtained at higher growth temperatures. The ALD-ZnO films grown at 150 °C were high quality with almost undetectable oxygen vacancy densities. Figure 5 presents the crystal structure analysis of ALD-ZnO films grown at different substrate temperatures. When the substrate temperature was raised from 70 °C to 125 °C, the profiles were almost unchanged, and the preferred crystal orientation was not particularly pronounced. It is worth mentioning that films prepared at temperatures as low as 70 °C also had good crystallinity. This result is consistent with the PL analysis and implies that the arrangement of the reaction atoms in the ALD process was nearly perfect, even at a very low deposition temperature. It can also be seen from Figure 5 that the ALD-ZnO film does not have a special preferential growth orientation and no second impurity phase is formed.

During gas sensing, electrons should transfer from the ZnO nanorods to the ZnO seed layer and then to the metal electrode. Hence, the quality of the ALD-ZnO seed layer is important, as it will affect the efficiency of the gas sensor. In this study, the average film thickness was about 100 nm, and the sheet resistances of ALD-ZnO films prepared at different deposition temperatures were determined with four-point probe measurements. The resistivity values were obtained from the sheet resistance and the film thickness, using the formula [29]:ρ = R_s_ × T
where ρ, R_s_, and T are resistivity, sheet resistance, and film thickness, respectively. It is generally believed that the semiconductor behavior of ZnO is controlled by point defects such as oxygen vacancies and zinc interstitials. Re-examination of the PL spectra of the ALD-ZnO films and comparison with the spectra of ZnO films prepared by other processes revealed that the point defect density in the ALD films was lower, indicated by the broadened peaks around 580 nm. The point defect concentration of the ALD-ZnO thin films gradually decreased as the growth temperature increased. At a growth temperature of 150 °C, the corresponding point defect peak in the PL spectrum has almost disappeared, and only the near-band-edge emission can be seen. Figure 6 shows that higher crystalline quality and lower defect density led to lower electrical resistivity as the substrate temperature was raised. Ultimately, a resistivity of 2 × 10^−3^ ohm·cm was reached at a growth temperature of 125 °C.

Figure 7 shows the result of growing a 100 nm ZnO seed layer by ALD, followed by hydrothermal growth of ZnO nanorods on the seed layer for 9 h. High-density ZnO nanorods with numerous micro-gaps between the rods facilitated the process of hydrogen sensing. Figure 8 presents the sensing results for different sensing temperature under the concertation of 10,000 ppm of hydrogen, using the ZnO nanorods prepared in this study. When sensing a reducing gas, the gas sensing sensitivity(S) can be expressed by the following relationship:S = Ra/Rg
where Rg represents the resistance measured in the reducing gas and Ra is the resistance measured in the reference gas of air [30,31].

ZnO is an n-type semiconductor. When it is exposed to air, O_2_ molecules are adsorbed on its surface, and electrons are taken from the surface of the ZnO nanostructures to form negative oxygen ions. When the O_2_ molecules capture electrons on the ZnO surface, a certain activation energy needs to be overcome to form the negative ions. As temperature accelerates this process, metal-oxide gas sensors (such as ZnO) require higher operating temperatures. The experimental results show that the sensitivity increased with higher sensing temperature, from 1.03 at 225 °C, to 1.32 at 380 °C after sensing 100 s in 10,000 ppm of hydrogen as shown in Figure 8. The hydrogen molecules reacted with the oxygen species adsorbed on the surface and reinjected the electrons onto the conduction band of the ZnO nanorods, thereby raising the current of the ZnO nanorods and increasing the sensitivity of the gas sensor. When the hydrogen gas is sensed, the data extractor has a fixed voltage of 5 volts applied on the electrodes. Therefore, the measured current value can be converted into a resistance according to Ohm’s law to facilitate the calculation of sensitivity by the formula shown above. The ALD-ZnO seed layer not only assisted with the growth of ZnO nanorods, but also helped facilitate the transfer of electrons to the external electrode, resulting in a more sensitive gas sensor. Therefore, the ZnO seed layer with high crystal quality, high density and high flatness is of great help to improve the sensitivity of the ZnO hydrogen sensor.

In addition to high sensitivity, an effective sensor needs good stability, meaning the sensor maintains its sensing ability over long periods of time. Figure 9 shows the sensitivity of the ZnO nanorod gas sensors after repeated exposures to 10,000 ppm of hydrogen. While the magnitude of the sensitivity remained stable to an average value of 1.2. Some current drift was detected in the process of gas sensing. This may be due to the fact that those sensing materials were prepared at much lower temperatures as compared to the sensing temperature (i.e., at 380 °C for sensing vs. 125 °C for ALD, and 90 °C for hydrothermal growth). At 380 °C, it seems that the sensing material is being annealed. The slight adjustment of the microstructures of materials will cause slight changes in their electrical characteristics. Therefore, the detected current (or resistance) in sensing material will also produce corresponding minor changes. It can be seen from Figure 9 that the sensitivity varies somewhat with the number of repeated measurements. However, the value of the variation is small, and the average value is maintained at about 1.2. This result represents that this gas sensor prepared in this study has good reproducibility and stability.

## 4. Conclusions

Preparation of ZnO seed layers by ALD yielded a material with good crystal quality and low resistance. The ALD-ZnO seed layers not only help overcome the energy barrier to grow ZnO nanorods but also promote electron transport when performing hydrogen sensing. The ZnO nanorods grown on ALD-ZnO seed layer effectively detected hydrogen with sensitivity ranging from 1.03 at 225 °C to 1.32 at 380 °C in 10,000 ppm of hydrogen.

## Figures and Tables

**Figure 1 micromachines-10-00491-f001:**
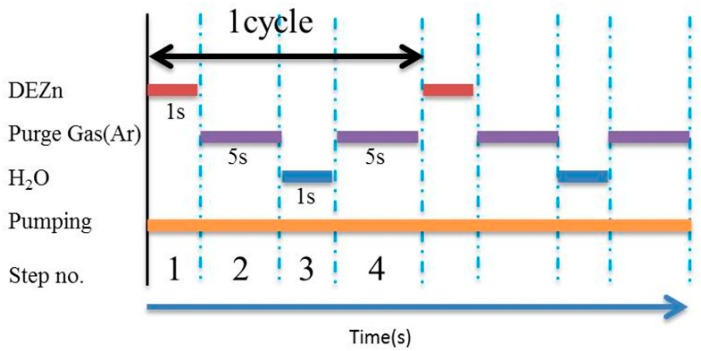
Sequence and time diagram for the deposition of ALD-ZnO seed layers.

**Figure 2 micromachines-10-00491-f002:**
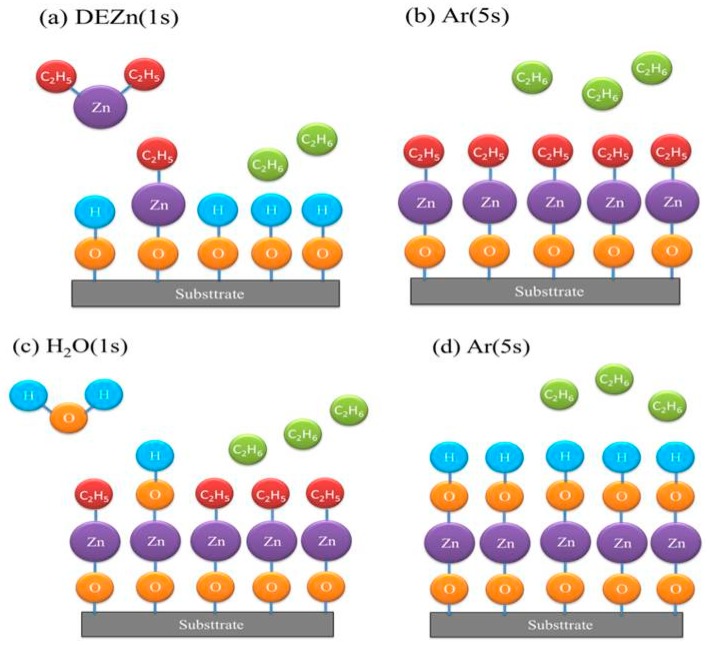
Schematic representation of the growth of ALD-ZnO film.

**Figure 3 micromachines-10-00491-f003:**
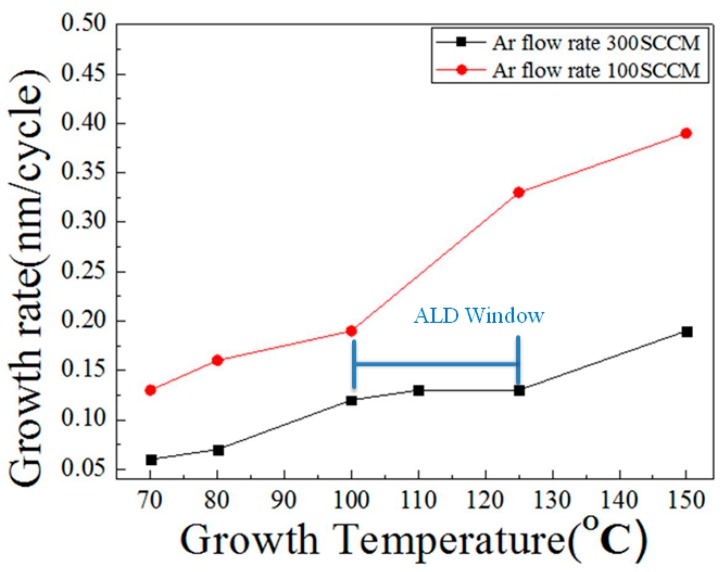
The growth window of atomic layer deposition (ALD).

**Figure 4 micromachines-10-00491-f004:**
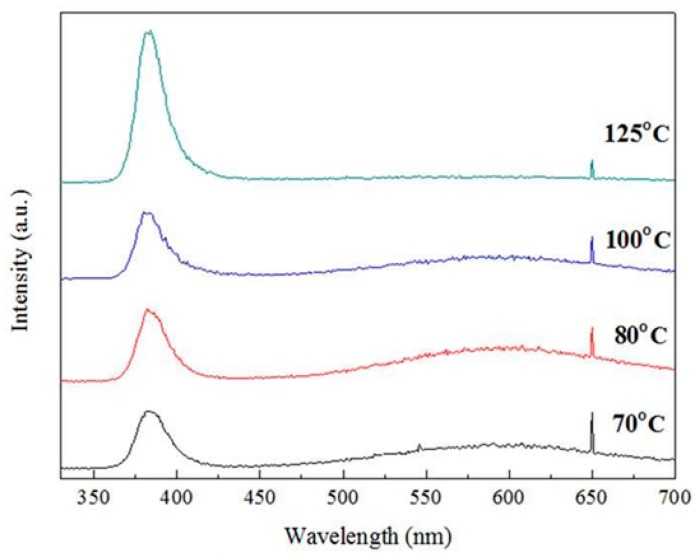
PL spectra of ALD-ZnO films.

**Figure 5 micromachines-10-00491-f005:**
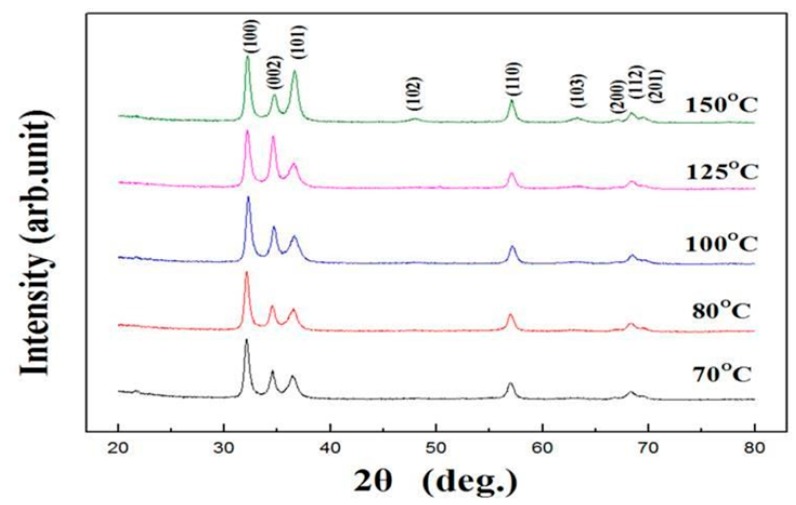
XRD patterns of ALD-ZnO prepared at different temperatures.

**Figure 6 micromachines-10-00491-f006:**
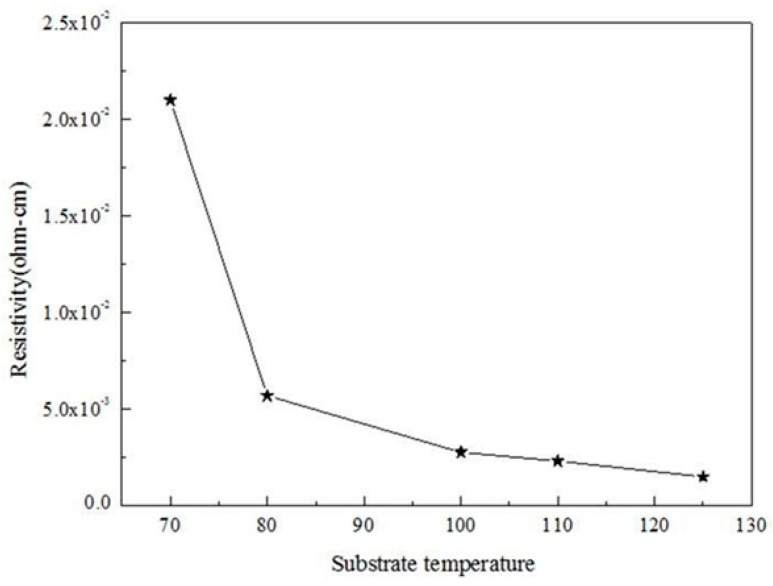
Resistivity of ALD-ZnO seed layers prepared at different deposition temperatures.

**Figure 7 micromachines-10-00491-f007:**
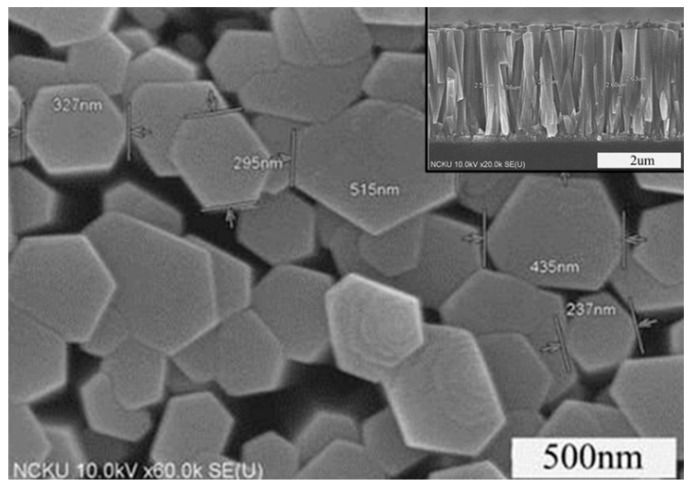
Top and cross-sectional views of ZnO nano-arrays.

**Figure 8 micromachines-10-00491-f008:**
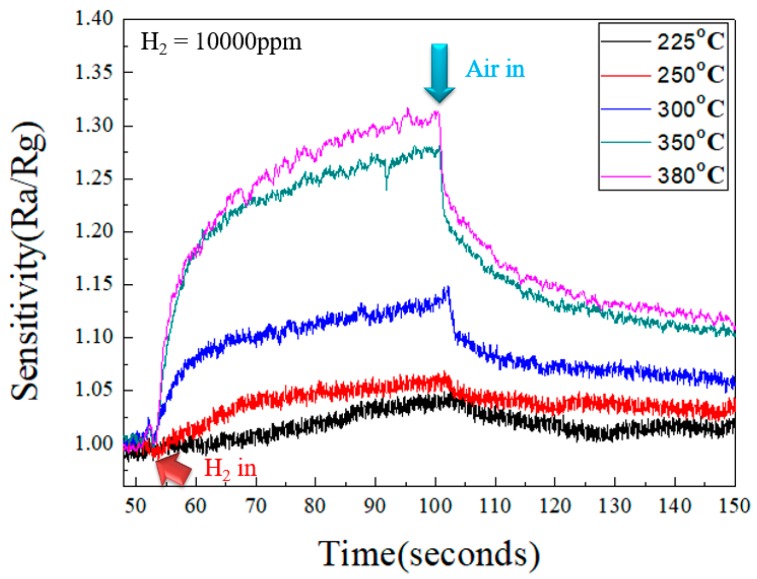
Sensitivity versus time of ZnO nanorods exposed to 10,000 ppm H_2_ at various operating temperatures.

**Figure 9 micromachines-10-00491-f009:**
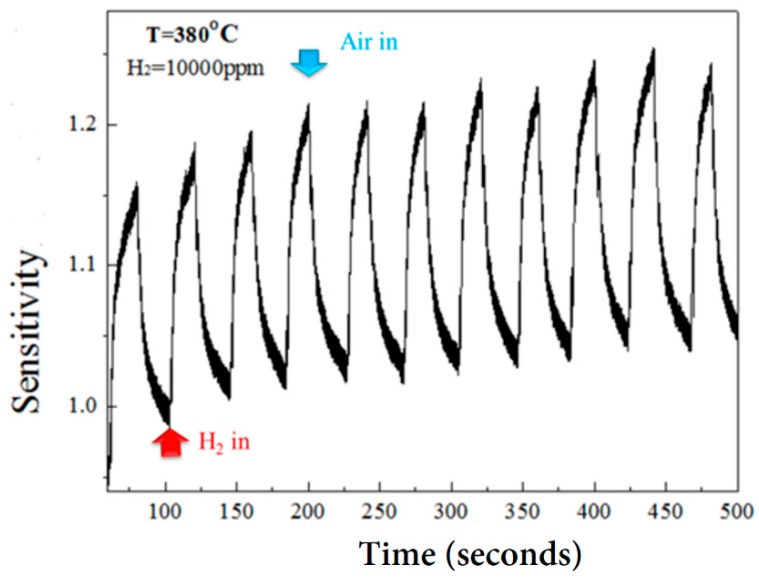
Sensitivity of ZnO nanorod sensor to repeated exposures of 10,000 ppm H_2_.

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
