# Peer review of "The Role of ALD-ZnO Seed Layers in the Growth of ZnO Nanorods for Hydrogen Sensing"

_micromachines, 2019, doi:10.3390/mi10070491_

Round 1

Reviewer 1 Report

The paper is interesting, however, the presented results are rather for a higher hydrogen concentrations:  10000 ppm, 150000? ppm in air - pls change to %.

The defined sensitivity S is rather the sensor response, than the sensitivity - in the v.146 is the term responses and in others sensitivity.

I think the authors should use the term response or relative response.

In Fig.9 the applied times of hydrogen dosing and air regenerations should be higher to better observe the stable signal of the sensor.

The applied operating temperatures are relatively high ~380 C - and it is not good from the point of view of power consumption- pls.comment.

The modern ALD ZnO technology is a major point of the paper.

The discussion of the relationship between the electrical conductivity, point defects, and crystal quality of ALD-ZnO films towards hydrogen detection should be extended in conclusions.

Author Response

Thank you for your comments and suggestions.They are good and constructive. 

Question 1: The paper is interesting, however, the presented results are rather for a higher hydrogen concentrations:  10000 ppm, 150000? ppm in air - pls change to %.

Response 1:

According to the theoretical calculation, the hydrogen explosion limit is 4.0% to 75.6% (volume concentration), which means that if the volume concentration of hydrogen in the air is between 4.0% and 75.6%, the fire source will explode. When the hydrogen concentration is less than 4.0% or greater than 75.6%, it will not explode even if it encounters a fire source. Converting concentrations of 10,000 ppm and 150,000 ppm into volume percent concentrations are equivalent to 1% and 15% respectively.

Question 2:The defined sensitivity S is rather the sensor response, than the sensitivity - in the v.146 is the term responses and in others sensitivity.

I think the authors should use the term response or relative response.

Response 2:

  The word "RESPONSE" is only a word problem in expression. In the field of gas sensing, it is generally expressed in terms of sensitivity.

Question 3:In Fig.9 the applied times of hydrogen dosing and air regenerations should be higher to better observe the stable signal of the sensor.

Response 3:

  We just took the data display repeated in 500 seconds. In fact, after a few days, the devices were measured again, and the reproducibility was still good. In fact, this is the characteristic of the oxide gas sensor, which will be quite stable in the air. The authors believe that repeating such measurements in 500 seconds is sufficient to demonstrate the stability of this device, which is similar to the general data shown in the journal paper for gas sensor.

Question 4:The applied operating temperatures are relatively high ~380 C - and it is not good from the point of view of power consumption- pls.comment.

Response 4:

Although the stability of oxide gas sensor in air is quite good, it generally works at high temperatures. Fortunately, it only needs to heat a small area of sensing nano material, and the energy provided does not need too much. So there is not much power consumed.

Question 5:The modern ALD ZnO technology is a major point of the paper.

Response 5:

There are many ways to prepare the seed layer of ZnO, including sputtering, thermal evaporation, sol-gel method, etc. The ZnO seed layer prepared by ALD method has excellent crystallinity, less crystal defects, high density and very flat surface. Therefore, the seed layer prepared by ALD-ZnO for ZnO nanostructure growth is expected to improve the gas sensing sensitivity.

Question 6:The discussion of the relationship between the electrical conductivity, point defects, and crystal quality of ALD-ZnO films towards hydrogen detection should be extended in conclusions.

Response 6:

  I have rewritten the conclusion section to more closely relate the relationship between ALD-ZnO seed layer and hydrogen sensing sensitivity as shown below.

"4. Conclusion

Preparation of ZnO seed layers by ALD yielded a material with good crystal quality and low resistance. The ALD-ZnO seed layers not only help overcome the energy barrier to grow ZnO nanorods but also promote electron transport when performing hydrogen sensing. The ZnO nanorods grown on ALD-ZnO seed layer effectively detected hydrogen with sensitivity ranging from 1.03 at 225oC to 1.32 at 380oC in 10000ppm of hydrogen."

Reviewer 2 Report

The paper presents an interesting technological advancement in the relevant field and fits the scope of the Journal. 

Overall, it is presented in an intelligible way, with a good quality of English language and grammar. 

I would suggest a few minor tips to further improve it:

- The Abstract should be rewritten in its initial part, stating a little better the background and aims

- Results and Discussions are difficult to follow. I would suggest making this section more schematic, when possible

- Limitations should be acknowledged

- A stronger take-home message is desired

In addition, few typos are present throughout the manuscript. 

Author Response

Comments and suggestions for authors:

The paper presents an interesting technological advancement in the relevant field and fits the scope of the Journal. 

Overall, it is presented in an intelligible way, with a good quality of English language and grammar. 

I would suggest a few minor tips to further improve it:

- The Abstract should be rewritten in its initial part, stating a little better the background and aims

- Results and Discussions are difficult to follow. I would suggest making this section more schematic, when possible

- Limitations should be acknowledged

- A stronger take-home message is desired

In addition, few typos are present throughout the manuscript. 

Response:

 Thank you for your comments and suggestions.They are good and constructive. 

 The background description and study purpose  have been added to the abstract section. e. It is also rewritten so that the outline of the entire paper can be completely depicted in abstract.In the results and discussion section, I also added some descriptions of theories and mechanisms according to your suggestion.

  I have modified the entire paper to make the results and discussion more clear and relevant.

Reviewer 3 Report

The manuscript by Lu et al reported the ALD based seed layer deposition and successive hydrothermal growth of ZnO nanorods for hydrogen sensing application. Structural, optical and morphological studies were evaluated using various characterization techniques.The manuscript presented shown no novelty and poorly written. Hence, I don’t recommend for publication in Micromachines. Following are the suggestions to improve the quality of the manuscript. 

The introduction is poor. Needs to be improved with various comparison of current literature sources. Already various ZnO based hydrogen sensors have developed. Novelty of this work should be clearly stated.

Support the novelty with proper references in the introduction part

More detailed experimental details is required.

Required more analytical results to support the doping effect like FESEM, XPS and HRTEM for all the samples.

Authors should dig out more information from XRD and PL.

What is the thickness of the thin films?

Sensing mechanism should be explained more technically.

How the sensor selectivity is proved?

English must be revised throughout the manuscript.

Author Response

Q1:The manuscript by Lu et al reported the ALD based seed layer deposition and successive hydrothermal growth of ZnO nanorods for hydrogen sensing application. Structural, optical and morphological studies were evaluated using various characterization techniques.The manuscript presented shown no novelty and poorly written. Hence, I don’t recommend for publication in Micromachines. Following are the suggestions to improve the quality of the manuscript. 

Response 1:

The overall structural issues of this paper vary from one reviewer to the other.Anyway, I will try my best to refer to your suggestions and comments for your approval.

Q2:

The introduction is poor. Needs to be improved with various comparison of current literature sources. Already various ZnO based hydrogen sensors have developed. Novelty of this work should be clearly stated.

Support the novelty with proper references in the introduction part

Response 2:

We have added some descriptions in the INTRODUCTION section as follows.

"S.Basu and A.Dutta developed two kinds of sensor structures, viz., a Pd/ZnO/p-Si heterojunction and Pd/ZnO/Zn metal-active insulator-metal (MIM) for hydrogen sensor applications. These two structures mainly rely on the hydrogen sensing properties of the Pd precious metal itself, not the individual contribution of the ZnO material. Sukon Phanichphant once reviewed the use of oxide semiconductors as a hydrogen sensor in 2014. From his review, it can be found that almost all of the materials are based on WO3 materials and all need to add precious metal Pt or Pd to get good results. Without the help of these precious metals, it is necessary to detect hydrogen at temperatures above 200-300oC or more.

In our research, cheap, non-toxic and abundant zinc oxide materials were used. It can detect low concentration of hydrogen without the help of expensive rare metals. The most important reason for this is that we had prepared the structure of the zinc oxide nano-rods and improved the material properties and electrical properties of the interface material. In other words, the ALD-ZnO seed layer was used to grow the ZnO nano-rods structure in this study."

Q3:More detailed experimental details is required.

Response 3:

We have added some descriptions in the experimental section as follows.

"The overall growth of ALD-ZnO cycle is mainly divided into four steps. Firstly, DEZn is introduced, which reacts with the hydroxyl group on the surface of the substrate to form a zinc-oxygen bond and produces an ethane(C2H6) by-product. In the second step, argon (Ar) is introduced to carry away the excess ethane(C2H6) by-product molecules and unreacted DEZn molecules. In the third step, water molecules are introduced to provide hydroxyls to bond zinc atoms and some ethane(C2H6) by-products are formed. In the fourth step, argon (Ar) is again introduced to carry away residual water molecules and excess ethane (C2H6) by-products molecules. The times to inject DEZn, argon, water and argon in sequence are set to 1 second, 5 seconds, 1 second, and 5 seconds on an automatic controller."

Q4:Required more analytical results to support the doping effect like FESEM, XPS and HRTEM for all the samples.

Response 4:

In this study, we did not dope ZnO. Hence it is not necessary to do the above analysis in this paper.The available data are sufficient to support the topics we want to discuss.

Q5: Authors should dig out more information from XRD and PL.

Response 5:

We have discussed the relationship between PL spectra and crystal quality ,structural defects ,emission spectra and energy gaps.We have added the analysis of XRD results as follows.

"It can also be seen from Fig. 5 that the ALD-ZnO film does not have a special preferential growth orientation and no second impurity phase is formed."

Q6:What is the thickness of the thin films?

Response 6:

The thickness of  ALD-ZnO films are about 100nm.

Q7:Sensing mechanism should be explained more technically.

Response 7:

We have added the following description in the INTROCUCTION section.

"The gas sensing mechanism can be illustrated by taking the N-type semiconductor material, ZnO, as an example. When the ZnO gas sensing material is placed under normal atmospheric conditions, O2 in the air will adsorb to the surface of ZnO. The adsorbed O2 will extract an electron from ZnO surface and transform itself into an O2- state, resulting in a decrease in electron density on the ZnO surface and an increase in resistance. When slightly heated, it accelerates the adsorption of O2 on the surface of ZnO to form a more stable adsorption state. If the temperature is higher than 150 °C, the O2- will convert to an O- adsorption state with electron. When the N-type semiconductor material is placed in an atmosphere of a reducing gas, the O- originally adsorbed on the ZnO surface will react with the reducing gas and release electrons back to the ZnO. This will increase the electron density of the ZnO and lead to a decrease in the resistance of it."

More detailed information about gas sensing mechanism suggested to refer the review paper published in Materials science &Engineering B 229(2018)206-217 by Ananya Dey.

Q8:How the sensor selectivity is proved?

Based on the limitations of the measurement equipment in this study, we only simulate the situation in the air.

Q9:English must be revised throughout the manuscript.

English is not my native language, so I can only express my meaning as simple and clear as possible.In fact, I have already paid for the person who has specially revised the article in the English-speaking country.I think the English and sentence patterns used in engineering and scientific papers should be as concise as possible. Because large part of its readers are not in English-speaking countries.

Round 2

Reviewer 3 Report

The revised manuscript is improved well. The authors have done the mentioned comments. The manuscript may be accepted in its present form. 

This manuscript is a resubmission of an earlier submission. The following is a list of the peer review reports and author responses from that submission.

Round 1

Reviewer 1 Report

According my opinion, the manuscript must be completely rewritten before it can be published. The authors should set the manuscript differently. First of all, they should write the abstract too carefully. In the introduction, the authors try to explain why ZnO is better than SnO2 as sensing metal oxide, but there is no reference to other works about of hydrogen sensors. Also giving an overview of the existing literature dealing with hydrogen sensors based on metal oxides could give the right frame to this manuscript. Furthermore, the work is unbalanced since there is a long description of the material and how to evaluate the degree of material crystallinity, but the sensor characterization is much too poor. Authors should describe more thoroughly the device and try to extract more information on performance, investigating the sensing mechanism. At the end, they should be give the comparison with literature works.

Author Response

I have rewritten the manuscript according to reviewer’s comments and suggestions.

The abstract is rewritten. Other works about of hydrogen sensors could be found in references[1-6].This paper is focus on the role of ALD-ZnO seed layer in ZnO sensor, overview can be easily found in my references. The sensor characterization is already described in paragraph of results and discussion:” Zinc oxide is an n-type semiconductor. When exposed to an air environment, oxygen molecules (O2) are adsorbed on the surface of the zinc oxide, and electrons are taken from the surface of the zinc oxide nanostructures to form a state of negative oxygen ions. When oxygen molecules capture electrons on the surface of zinc oxide, a certain activation energy needs to be overcome to form negative ions. As temperature accelerates this process, metal oxide gas sensors (such as ZnO) need to be operated at enhanced operating temperatures.” Almost all oxide gas sensors have similar sensing mechanisms.More mechanisms can be found in references[ 1-2] or other text books. The mechanism of oxide gas sensor had been well developed.

Reviewer 2 Report

Dear Authors,

your proposal "The role of ALD-ZnO film in the preparation of
the nano ZnO hydrogen sensor" has to be revised before topass
to major revision. You need to reorganize the figures in the
text (the current version has many figures at page 4 and 5).
It is important to define the sensitivity in a dedicated
equation and not in the text. I didn't understand the link
between the sensitivity defined in #121 and the  Igas/Iair
(figure 8). It seems that you considered Rg as the baseline
resistance, but usually the literature uses the resistance
under air or N2 as the reference. Furthermore, the
measurement chamber is missing: i.e. did you use a heater or
a oven for reaching 380degreesCelsius?What is the flux of
Hydrogen?The volume of the chamber?The conclusions are poor:
I'd appreciate some interesting application of this method in
industrial field.

Check for the following comments:

- #7: the acronym of ALD is missing (Atomic..);
- #13: the last sentence is not clear. You need to underline the method and the factors that play a crucial role in the sensor performance;
- #33-#34: you summarized the refences [10-17] at the end of
the sentence. Please, distribute the references in the sentence;
- #39: needing to define ALD acronym in the row #7;
- #46: thickness of Si substrate?
- #50: add the acronym "(SEM)";
- #59: space between "...window" and "Fig.3 ...". Check in
all document;
- #60: check for the symbol of degrees;
- #89: needing space before "Hense";
- #89: meaning of "impoerant";
- there is no reference in the text of equation (1);
- figure 6: the labels and the values on y-axis is not
readable;
- figure 7: 2um in the top left corner is not so readable;
- figure 8: the measure unit is "s" and not "sec".

Best regards.

Author Response

Please find our responses in the attachment.

Reviewer 3 Report

The paper deals with the deposition of ZnO films with a columnar nanorod morphology, which is beneficial for gas sensing, on silicon substrates. The authors show that high-Quality material can be obtained via a facil hydrothermal route when the hydrothermal process is performed on ZnO seed layers deposited by the atomic layer deposition (ALD) technique. Such low-temperature depsoition techniques are of high interst when gas sensing materials are to be deposited on silicon microheaters. As a proof of the favorable gas sensing properties H2 gas sensing tests have been performed. 

Before publication improvements in the language should be implemented to enhance the Level of readability. Also some technical issues should be clarified. Detaialed comments and suggestions are in the attached file.

Author Response

I have rewritten the manuscript according to reviewer3’s comments and suggestions.

They are very good suggestions. Thank you for your effort to let the paper has a more readability.

Round 2

Reviewer 1 Report

Your revised paper is not yet well organized and is not clearly rewritten as indicated below. Therefore, I regrettably have to get Major Revision. Major technical revision should be made thorough examination of my comments.

·         There are several mistyping errors that indicate a little care in the revision of the text. I refer to the lines 29, 44, 69, 73, 83, 153. The sentence of line 59 is ungrammatical.

·         The acronyms must always be explained, also for the XRD.

·         Fig. 8 doesn’t show the title on the x axis.

·         It’ wrong to report the sensitivity, this is calculated from the calibration curve. Your measure is device response.

·         Again, the description of the degree of crystallinity of the material should be streamlined and more attention should be paid to the usefulness of the material, with a comparison with the literature works.

Reviewer 2 Report

Dear Authors,

your proposal "The role of ALD-ZnO film in the preparation
of the nano ZnO hydrogen sensor" has to be revised again.
The abstract had to be implement by adding a brief
introduction: now, it is diffucult to understand the
narrative of the paper. There are many typographical errors
(e.g. #21 double parenthesis or the equation indexes are
missing and many others).
There is no trace of the measurement chamber for H2 target
gas (plus the volume of chamber and the flow rate) and the
heater for reaching 380degrees Celsius.
The Internal System of Units for seconds is "s" and not
"normally sec".

Best regards.

Reviewer 3 Report

the paper is OK, provided some small additional amendments (GREEN) are implemented.
